# High RRM2 expression has poor prognosis in specific types of breast cancer

**Shen-chao Shi[1], Yi Zhang[2]☯\*, Tao Wang[3]☯\***

**1** Department of Hepatobiliary Surgery, Union Hospital, Tongji Medical College, Huazhong University of Science and Technology, Wuhan, China, **2** Department of Endocrinology, Hubei No. 3 People's Hospital of Jianghan University, Wuhan, China, **3** Department of Thyroid and Breast Surgery, Hubei No. 3 People's Hospital of Jianghan University, Wuhan, China

☯ These authors contributed equally to this work.
\* zhangy_zsyy@126.com (YZ); drwontao@gmail.com (TW)

## Abstract

### Background

RRM2 plays an important role in different malignant tumors, but there are few studies in breast cancer. Public databases were used to analyze the expression of RRM2 in breast cancer and its prognostic value.

### Materials and methods

A total of 2,509 breast cancer samples were downloaded from the METABRIC database. The relationship between RRM2 expression and clinical pathology was evaluated. Using the BCIP database and real-time-PCR, and western blotting, RRM2 mRNA and protein expression of RRM2 in breast cancer tissues and cell lines were evaluated. Univariate and multivariate analysis defined independent prognostic factors that affected the overall survival of patients with breast cancer. The Kaplan-Meier method was used to study the relationship between the high expression of RRM2 and overall survival and distant metastasis-free survival (DMFS) of breast cancer patients. Finally, We performed Gene Set Enrichment Analysis (GSEA) and obtained the relevant pathways associated with high expression of RRM2 potentially influencing breast cancer progression.

### Results

RRM2 expression was significantly correlated with age, tumor size, grade, menopausal status, molecular typing, ER, PR, and Her-2 of patients with breast cancer(P<0.05). Univariate and multivariate regression analysis showed that RRM2, the number of positive lymph nodes, ER, Her-2, tumor size, and tumor stage can be used as independent prognostic factors for overall survival of patients with breast cancer. Kaplan-Meier analysis showed that in patients with Luminal A and Normal like breast cancers and Stage1 and stage2 breast cancers, patients with high expression of RRM2 had worse overall survival and DMFS. The analysis of the GSEA pathway showed that RRM2 is mainly enriched in the ERBB signaling pathway and other pathways.

**Data Availability Statement:** All relevant data are within the paper and its Supporting Information files. The Distant metastasis-free survival (DMFS) online analysis data were obtained through the website https://kmplot.com/analysis/. Breasts

Cancer Integrative Platform (BCIP) data obtained through http://www.omicsnet.org/bcancer/ database.

**Funding:** This work was supported by National Natural Science Foundation of China (81771718).

**Competing interests:** The authors have declared that no competing interests exist.

## Conclusion

The high expression of RRM2 has a worse prognosis in patients with breast cancer with specific features. It can be used as a biomarker for the prognosis of breast cancer.

## Introduction

Breast cancer has become the leading cause of global cancer in 2020, accounting for 11.7% of all new cases of malignant tumors. Among female patients with malignant tumors, the incidence rate is 24.5%, and the mortality rate is 15.5%, both ranking first among cancers [1]. Aibonucleotide reductase (RR) can catalyze the conversion of nucleoside diphosphate to deoxynucleoside diphosphate, which is the rate-limiting enzyme for DNA synthesis and repair, and plays an important role in cell division, proliferation and differentiation. RR is composed of ribonucleotide reductase M1 (RRM1), ribonucleotide reductase M2(RRM2), and the newly discovered p53R2 subunit. The activity of the holoenzyme is determined by the expression of the RRM2 protein [2, 3].

Studies have shown that gene mutations and gene disorders are the main reasons leading to the occurrence and development of many cancers [4]. The differential expression of RRM2 is closely related to the occurrence, development, and prognosis of pancreatic cancer, gastric cancer, non-small cell lung cancer, liver cancer, malignant endometriosis, and other tumors [5–10]. Although targeted therapies, immunotherapy, chemotherapy, and radiotherapy are effective in the treatment of breast cancer, morbidity and mortality are still increasing. Therefore, it is very important to find new biomarkers and therapeutic targets that affect the survival and prognosis of breast cancer.

To evaluate the role of RRM2 as a prognostic biomarker in breast cancer, we analyzed the relationship between the expression level of RRM2 in the METABRIC database and the overall survival and DMFS of patients with breast cancer. First, 1350 breast cancer samples were downloaded and sorted, and then SPSS statistical software was used to analyze the relationship between the expression level of RRM2 and various clinicopathological characteristics. The level of RRM2 mRNA and protein expression in breast cancer tissues and cell lines was then analyzed using the BCIP database, qRT-PCR, western blotting. Univariate and multivariate Cox regression analysis obtained independent prognostic factors that affect the overall survival of breast cancer. The Kaplan-Meier (KM) method was used to explore the relationship between high expression of RRM2 and overall survival and DMFS of breast cancer, and to study the effect of RRM2 expression on overall survival and DMFS in specific types of breast cancer. Finally, we defined pathways associated with the high expression of RRM2 and breast cancer progression through GSEA analysis.

## Materials and methods

### METABRIC database

The Molecular Taxonomy of Breast Cancer International Consortium (METABRIC) is a public database of breast cancer. We downloaded clinical data from breast cancer patients and the Z-score value of mRNA expression through the cBioportal website (http://www.cbioportal.org/) [11–13]. After filtering, 1350 breast cancer samples were used for subsequent analysis.

## BCIP database

The Breast cancer integrative platform (BCIP) is a website that enables the analysis and visualization profiles of breast cancer patients. The data for this platform derives from public databases, including METABRIC, The Cancer Genome Atlas (TCGA), and Gene Expression Omnibus (GEO) datasets. We obtained the mRNA expression of RRM2 in breast cancer tissues and adjacent tissues through BCIP.

## Clinical specimens

Breast cancer tissues and adjacent tissues were collected between January 2018 and August 2020 from 45 patients, who underwent surgery at the Department of Thyroid and Breast Surgery, Hubei No. 3 People's Hospital of Jianghan University, immediately stored in liquid nitrogen and stored at -80˚C. The present study was approved by the Ethics Committee of The Hubei No.3 People's Hospital of Jianghan University.

## Cell culture

The human mammary epithelial cell line MCF-10A, breast cancer cell lines MCF-7, BT474, MDA-MB-453, and MDA-MB-231 were obtained from the Culture Collection of the Chinese Academy of Sciences (Shanghai, China), and were cultured in Dulbecco's modified Eagle's medium (DMEM; Thermo Fisher Scientific; USA) supplemented with 10% fetal bovine serum (FBS; Gibco; USA), and penicillin/streptomycin at 37˚C in a humidified incubator with 5% $CO_2$ incubator.

## qRT-PCR

The cell lines required for this experiment were purchased from the Shanghai Cell Bank of the Chinese Academy of Sciences. Total RNA was extracted from breast epithelial cell line MCF-10A and breast cancer cell line MCF-7, BT474, MDA-MB-453, MDA-MB-231.Using the Bio-Rad Real-time fluorescent quantitative PCR instrument, GAPDH as an internal reference, adopt two-step PCR reaction program, conditions are predenaturation at 95˚C for 10 min; denaturation at 95˚C for 15 seconds, annealing/extension at 60˚C for 60 seconds, 42 cycles. By analyzing and deriving the Ct value of each sample, calculating the 2-△△Ct value, and finally determining the relative expression level of mRNA [14]. Primer sequence: GAPDH (Forward 5'-ATGGCACCGTCAAGGCTG-3' and Reverse 5'-AGCATCGCCCCACTTGATTT-3'); RRM2 (Forward 5'-CTGGCTCAAGAAACGAGGACT-3' and Reverse 5'-ACATCAGGCAAG CAAAATCACA-3').

## Western blotting

A total of 100 mg of tissue was finely cut and placed in a test tube, and an appropriate amount of pre-cooled PBS was added; the samples were centrifuged for 3 min. Next, the extraction reagent (Sangon Biotech, One Step Animal Tissue Active Protein Extraction Kit) was added to the test tube, and the sample was homogenized on ice and incubated for 20 minutes. The sample was then centrifuged at 12,000 rpm for 10 minutes, and the supernatant containing the total tissue protein was removed. The protein concentration was determined using the BCA protein detection kit (Tiangen Biotechnology, Beijing, China). The protein was denatured at 100˚C for 10 minutes and the samples were loaded and separated by 10% SDS-polyacrylamide gel electrophoresis (SDS-PAGE). The protein was then transferred to a polyvinylidene fluoride (PVDF) membrane (Millipore, Bedford, MA, USA). The membrane was blocked with TBST containing 5% skim milk for 1 hour and then incubated with the specific primary antibody

overnight at 4°C with shaking. The membrane was washed three times with TBST for 10 minutes each time and incubated with the secondary antibody for 1 hour at room temperature. Protein detection was carried out using ECL substrate (Pierce) and signal intensities were quantified by Image J. Antibodies (Abcam) were used at the following dilutions: RRM2 (1:3000) and GAPDH (1:5000). Secondary antibodies used were HRP goat anti-mouse and HRP goat anti-rabbit IgG antibodies (Proteintech).

## Kaplan–Meier survival analysis

The KM-plot database simultaneously integrates gene expression and clinical data. Survival curves were calculated using the "survival" package. The online KM-plot database can be used to identify the correlation of individual RRM2 mRNA expression in survival analysis, including examination of overall survival (OS), recurrence-free survival (RFS), distant metastasis-free survival (DMFS), and postprogression survival period (PPS).

## GSEA (Gene Set Enrichment Analysis)

We performed GSEA analysis to study the molecular mechanism of RRM2 on breast cancer [15]. The GSEA 4.0.3 software was downloaded from Broad Institute (http://www.gsea-msigdb.org/gsea/index.jsp). The median value of RRM2 expression was used to classify patients into high expression groups and low expression groups, where FDR<0.05; The number of permutations: 1000 times; and R software was used to analyze and draw a GSEA enrichment analysis chart.

## Statistics analysis

All statistical analyses were performed with R (R version 4.0.3) [16] and SPSS (version 22) statistical software. The median expression of mRNA was used to classify RRM2 as high or low expression [17, 18]. The relationship between the expression level of RRM2 and clinicopathological characteristics were statistically analyzed by chi-square test or Fisher exact test. The KM method with the log-rank test was used to analyze the OS of patients. The ROC curve was used to evaluate the diagnostic value of RRM2 gene expression, and the area under curve (AUC) represents the diagnostic value. Univariate cox analysis was used to screen potential prognostic factors, and multivariate Cox analysis was used to screen independent prognostic factors that affect the OS of breast cancer patients.

# Results

## The relationship between the expression of RRM2 and the clinicopathological characteristics in breast cancer patients

To study the relationship between RRM2 expression and clinicopathological characteristics, we downloaded breast cancer patient data from the METABRIC database, which contained 2509 breast cancer samples. After censored cases were removed, a total of 1350 breast cancer patient samples were obtained. Among the patients of breast cancer, there were 630 individuals aged <60 years and 719 individuals aged ≥60 years; 1228 individuals with tumor stage 1/2 and 120 individuals with tumor stage 3/4; 605 individuals with a tumor size ≤2cm, 684 individuals with a tumor size ≤5cm and 60 individuals with a tumor size >5cm; 113 individuals with Grade 1, 532 individuals with Grade 2 and 704 individuals with Grade 3; 312 individuals were premenopausal, and 1,037 individuals were postmenopausal. In total, 134 individuals had Basal-like subtype, 156 individuals had Claudin-low subtype, 137 individuals had Her-2 enriched subtype, 498 individuals had Luminal A, 331 individuals had Luminal B, 93

individuals had Normal-like breast cancer; 630 individuals with lymph node metastasis, and 719 individuals without lymph node metastasis. Overall, 313 individuals had ER negative breast cancer, and 1,036 individuals had ER positive; 647 individuals had PR negative and 702 individuals had PR positive; 1,181 individuals with Her-2 negative, and 168 individuals had Her-2 positive breast cancer. The statistical analysis revealed correlations between RRM2 expression and clinicopathological parameters. Our study showed that the expression of RRM2 significantly correlated with age (P<0.001), tumor size (P = 0.039), grade (P<0.001), menopausal status (P<0.001), molecular typing (P<0.001), ER (P<0.001), PR (P<0.001), and Her-2 status (P<0.001), there was no significant correlation between the number of positive lymph nodes (P = 0.194) and tumor stage (P = 0.087) (Table 1).

## RRM2 mRNA and protein expression were significantly up-regulated in breast cancer tissues and breast cancer cell lines

Through the BCIP database, we analyzed the expression level of RRM2 in 4 data sets (METAB-RIC, TCGA Agilent, TCGA RNA-seq, GSE5364), and the results showed that RRM2 expression in breast cancer tissues was significantly up-regulated compared to adjacent normal tissues (Fig 1). In the breast cancer cell lines MCF-7, MDA-MB-231, MDA-MB-453, BT-474, the relative expression level of RRM2 mRNA was significantly up-regulated, compared to the breast epithelial cell line MCF-10A (Fig 1). We then detected the relative mRNA expression of RRM2 in 45 breast cancers and their paired adjacent normal tissues by qRT-PCR. The qRT-PCR results showed that the relative expression of RRM2 mRNA was higher in breast cancer tissues than in adjacent normal tissues (Fig 2A). Next, we detected the protein expression of RRM2 by western blotting. The protein expression of RRM2 was significantly increased in breast cancer tissues compared to adjacent normal tissues. We also detected the level of protein expression of RRM2 in breast cancer cell lines, and the same trend was obtained (Fig 2B and 2C).

## Expression of RRM2 associated with different clinicopathological characteristics of breast cancer patients

As shown in Fig 3, using R software, the boxplots indicate that the expression of RRM2 was higher in the <60-year-old subgroup (P = 1.09e-08). The expression of RRM2 was higher in the Basal and Her-2 types, and lower in the Luminal A and Luminal B types (P = 3.46e-79). The expression of RRM2 was positively correlated with the grade of breast cancer grade (P = 1.57e-32). RRM2 expression was higher in the premenopausal subgroup (P = 2.44e-09); in the Her-2 positive patients (P = 1.11e-11); and in hormone receptor negative (ER-, PR-negative) subtypes (P = 1.03e-7; P = 4.68e-21). However, we observed that there was no significant correlation between RRM2 expression and tumor stage (P = 0.159), tumor size (P = 0.0141) or the number of positive lymph nodes (P = 0.361).

## Univariate and multivariate Cox regression analysis

To obtain independent prognostic factors that influence the OS of breast cancer, we performed a univariate and multivariate Cox regression analysis. In the univariate Cox regression analysis, RRM2, the number of positive lymph nodes, ER, PR, Her-2, Grade, Tumor size, Tumor stage have significant correlations with OS (P<0.001), (Fig 4). All the above factors with significant differences were selected for multivariate Cox regression analysis. Only RRM2 expression, the number of positive lymph nodes, ER, Her-2, tumor size, tumor stage are significantly correlated with OS (P<0.05), and could be used as independent prognostic factors for predicting OS of breast cancer patients (Fig 4).

**Table 1. The correlation between the expression level of RRM2 and the clinicopathological characteristics of breast cancer patients in the METABRIC database.**

| Clinicopathological | Whole Cohort (n = 1349) | | P value |
| --- | --- | --- | --- |
| | **RRM2 High n = 675** | **RRM2 Low n = 674** | |
| **Age at diagnosis (years)** | | | <0.001 |
| <60 | 355 | 275 | |
| ≥60 | 320 | 399 | |
| **Tumor stage** | | | 0.087 |
| 0 | 0 | 1 | |
| 1 | 210 | 246 | |
| 2 | 394 | 378 | |
| 3 | 65 | 46 | |
| 4 | 6 | 3 | |
| **Tumor size** | | | 0.039 |
| ≤2cm | 282 | 323 | |
| ≤5cm | 357 | 327 | |
| >5cm | 36 | 24 | |
| Grade | | | <0.001 |
| 1 | 27 | 86 | |
| 2 | 197 | 335 | |
| 3 | 451 | 253 | |
| **Menopausal state** | | | <0.001 |
| Pre- | 201 | 111 | |
| Post- | 474 | 563 | |
| **PAM50+Claudin** | | | <0.001 |
| Basal-like | 128 | 6 | |
| Claudin-low | 95 | 61 | |
| Her-2 | 110 | 27 | |
| Luminal A | 147 | 351 | |
| Luminal B | 145 | 186 | |
| Normal-like | 50 | 43 | |
| **Lymph Node Number** | | | 0.194 |
| 0 | 350 | 369 | |
| 1–3 | 208 | 212 | |
| ≥4 | 117 | 93 | |
| **ER state** | | | <0.001 |
| Negative | 275 | 38 | |
| Positive | 400 | 636 | |
| **PR state** | | | <0.001 |
| Negative | 396 | 251 | |
| Positive | 279 | 423 | |
| **Her-2 state** | | | <0.001 |
| Negative | 554 | 627 | |
| Positive | 121 | 47 | |

## High RRM2 expression was associated with poor overall survival in breast cancer patients

The longest follow-up time for patients was 351 months, and the median follow-up time was 116 months. OS was significantly different across breast cancer subtypes. The Luminal A

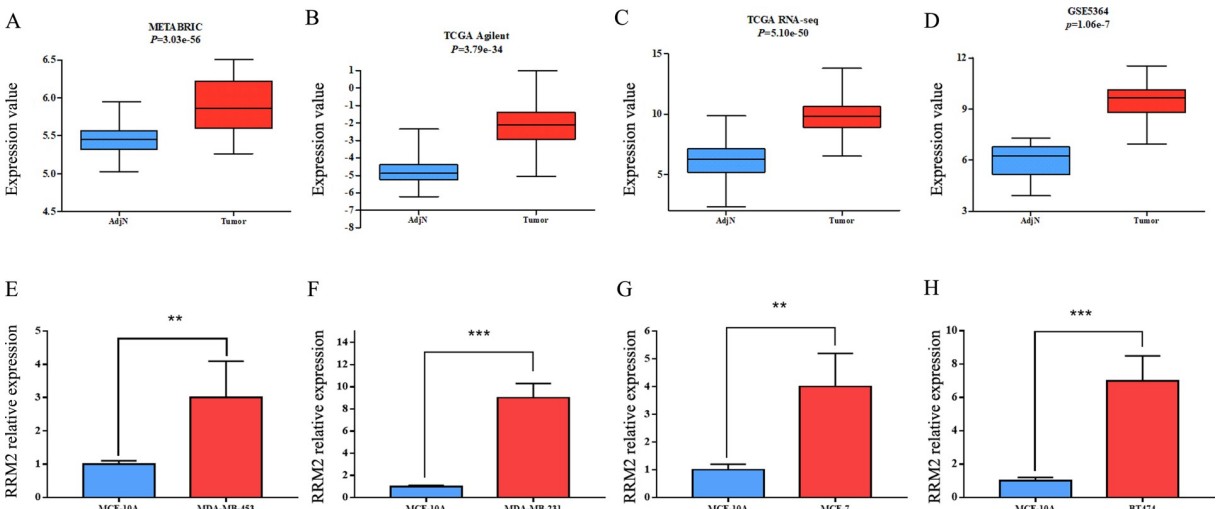

**Fig 1. Expression of RRM2 in breast cancer tissues from public databases and breast cancer cell lines by qRT-PCR.** (A) METABRIC; (B) TCGA Agilent; (C) TCGA RNA-seq; (D) GSE5364; (E) MDA-MB-453; (F) MDA-MB-231; (G) MCF-7; and (H) BT474.

subtype had the best prognosis, while Her-2 positive and basal-like subtypes had the worst prognosis (P<0.05) (Fig 5). In different tumor stages, the OS of the patients was significantly different (P<0.05). Stage 1 breast cancer patients had the best OS and stage 4 the worst (Fig 5). To evaluate the diagnostic value of RRM2, we used the KM method for analysis. The 1-, 3-, and 5-year AUC values were 0.703, 0.694, and 0.654, respectively. The ROC curve is shown in Fig 5. In all samples (n = 1350), the high RRM2 expression group had the worse OS (P<0.05) (Fig 5).

## The high RRM2 expression group had poor overall survival among Luminal A and Normal-like breast cancer patients

To further evaluate the impact of RRM2 on breast cancer patients, we evaluated the relationship between RRM2 expression levels and survival of breast cancer patients in different molecular subtypes. We found that the RRM2 high-expression group had poor OS in patients with the Luminal A subtype and the Normal-like subtype breast cancer patients (P<0.05). The expression of RRM2 did not has a significant correlation with the Basal-like subtype, the Luminal B subtype, Claudin low subtype, or Her-2 enriched subtype (Fig 6).

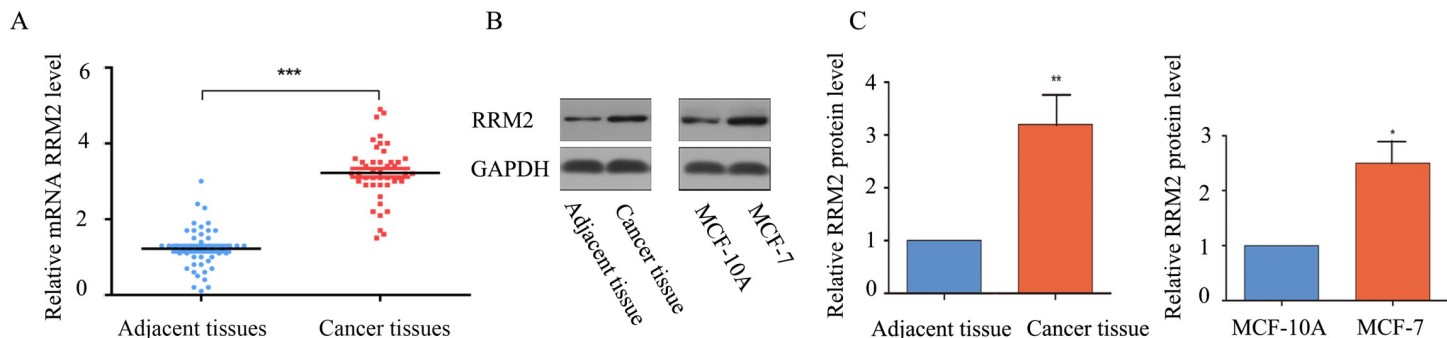

**Fig 2. The mRNA and protein expression levels of RRM2 in breast cancer.** (A) The relative expression level of RRM2 mRNA was higher in breast cancer tissues than in adjacent normal tissues. (B) The protein expression of RRM2 was higher in breast cancer tissues and breast cancer cell lines. (C) Quantitative results of protein expression levels obtained by Image J.

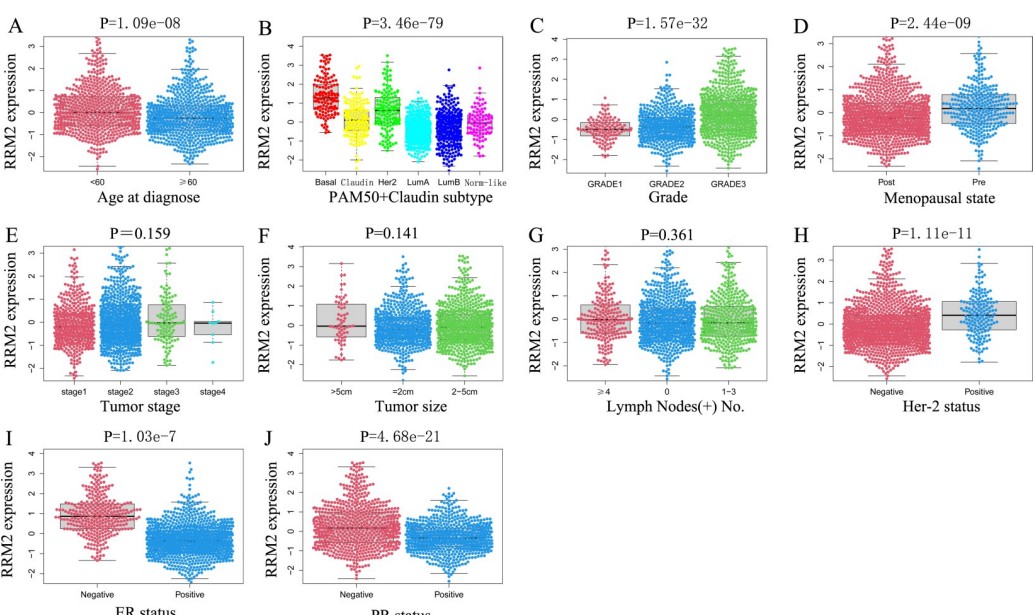

**Fig 3. The scatter plot shows the expression of RRM2 in the clinicopathological characteristics of METABRIC.** (A) Age; (B) Molecular subtype; (C) Grade; (D) Menopausal state; (E) Tumor stage; (F) Tumor size; (G) Lymph Node; (H) Her-2 status; (I) ER status; (J) PR status.

## Among stage1 and stage2 breast cancer patients, the RRM2 high expression group had poor overall survival

The OS of patients with breast cancer in different stages was analyzed and the results showed that the high expression of RRM2 was associated with poor OS in patients with stage1 and stage2 breast cancer(P<0.001). In stage3 breast cancer patients, the expression level of RRM2 was not significantly correlated with the OS of the patients (P = 0.065) (Fig 7). In the METABRIC database, due to the few patients in stage 0 and stage 4, these were not evaluated.

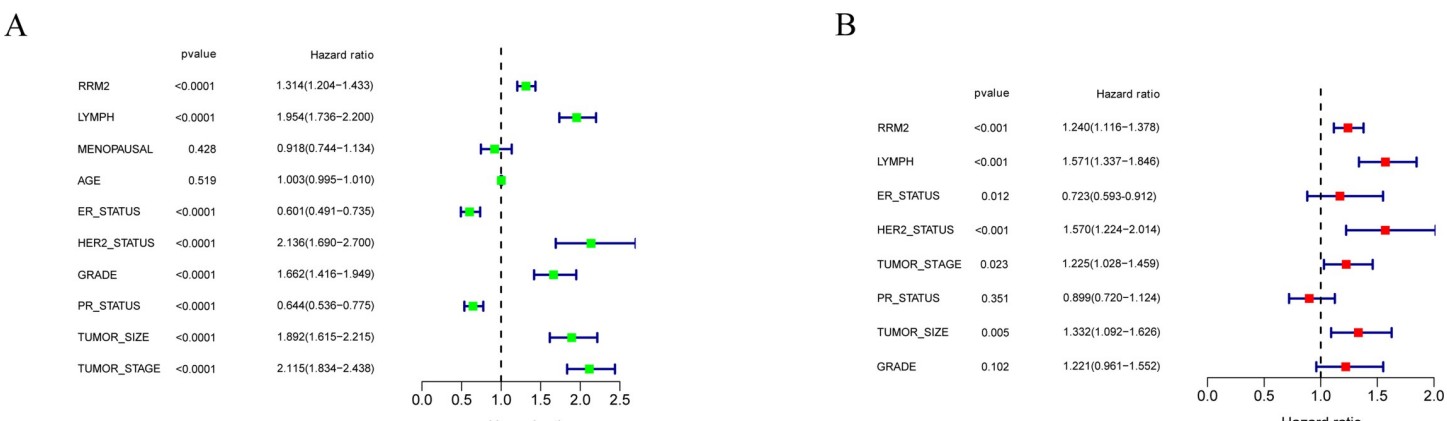

**Fig 4. Forest plots showing univariate and multivariate Cox regression analysis of the overall survival of breast cancer patients.** RRM2, the number of positive lymph nodes, ER status, Her-2 status, Tumor size, and Tumor stage were significantly correlated with overall survival (P<0.05). (A): Univariate Cox analysis. (B): Multivariate Cox analysis.

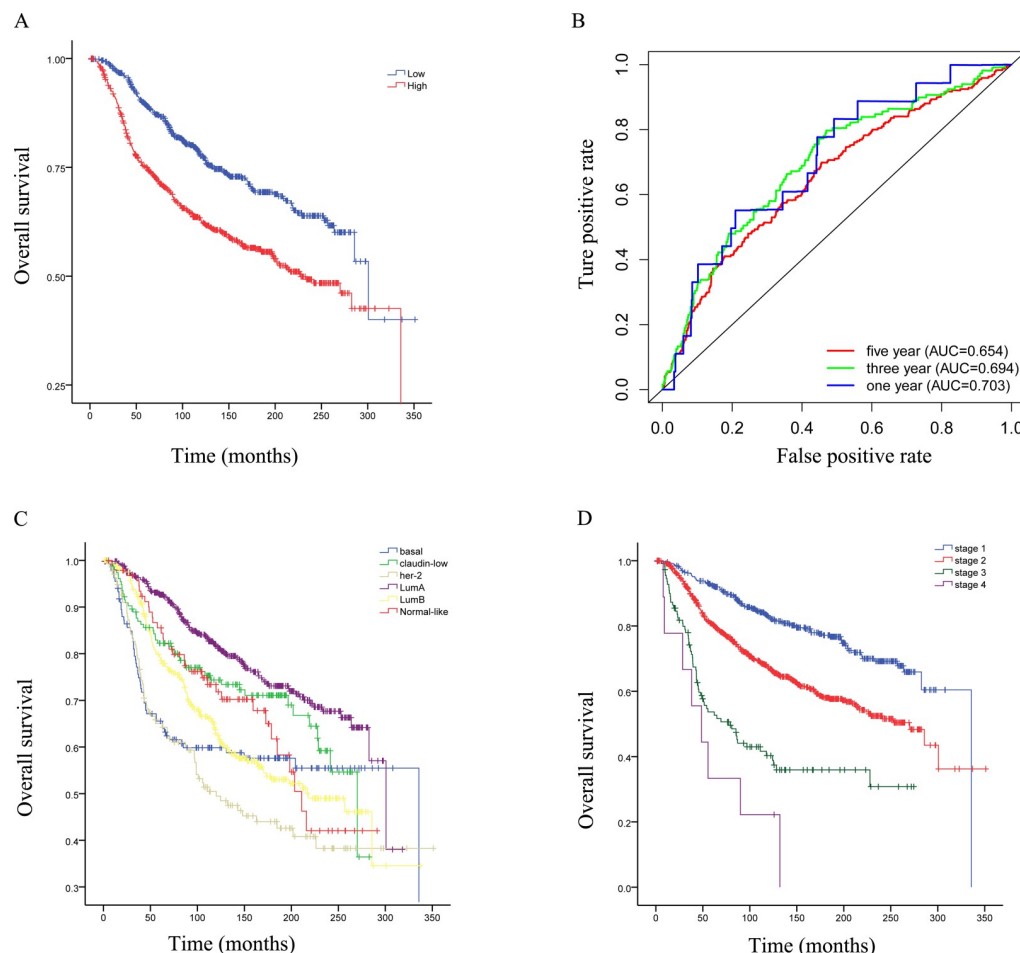

**Fig 5. Overall survival and ROC curve.** (A): Breast cancer patients with High-RRM2 expression have poor overall survival compared to those with Low-RRM2 expression. (P<0.05). (B): The ROC curve shows the diagnostic value of RRM2 for patients with breast cancer. (C): Overall survival analysis showed that patients with the Luminal A subtype of breast cancer have the best prognosis, and the Basal-like subtype has the worst prognosis in 1,350 breast cancer patients. (D): The high expression of RRM2 has poor overall survival in stage 1 and stage 2 breast cancer patients. (P<0.05).

## The high RRM2 expression group had poor distant metastasis-free survival (DMFS) in Luminal A breast cancer patients

To evaluate the effects of RRM2 expression on DMFS in breast cancer patients, we performed KM online survival analysis. Among all the breast cancer patients, the RRM2 high expression group had worse DMFS (P < 0.05). We further evaluated the effects of RRM2 expression on DMFS in patients with different subtypes of breast cancer, and the results showed that in Luminal A breast cancer patients, high expression of RRM2 had worse DMFS (P < 0.05). However, the expression of RRM2 was not significantly correlated with the Basal-like subtype, the Luminal B subtype, basal like subtype, or the Her-2 enriched subtype, as shown in Fig 8).

## Gene Set Enrichment Analysis (GSEA)

The analysis of the RRM2 KEGG pathway ex by GSEA software showed that the high RRM2 expression group was mainly enriched in the ERBB signaling pathway, the JAK-STAT signaling pathway, the mTOR signaling pathway, the P53 signaling pathway, the VEGF signaling

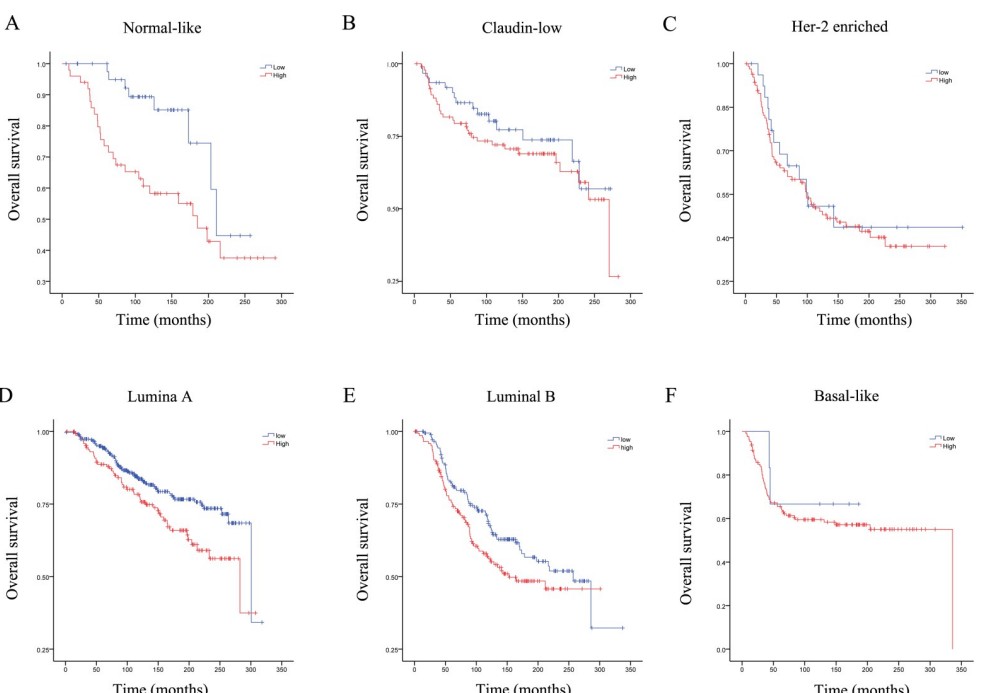

**Fig 6. The effect of RRM2 expression on the overall survival of breast cancer patients with different molecular subtypes.** We found that in Luminal A and Normal like breast cancers, the High RRM2 expression group had poor overall survival. (A) Normal-like; (B) Claudin-low; (C) Her-2 enriched; (D) Luminal A; (E) Luminal B; (F) Basal-like.

pathway, the WNT signaling pathway (Fig 9). These pathways play an important role in the development of a variety of malignant tumors.

## Discussion

The expression and role of RRM2 have been reported in a variety of cancers [10, 19]. However, its role in breast cancer and its mechanism are still unclear. Klimaszewska-Wiśniewska A et al.

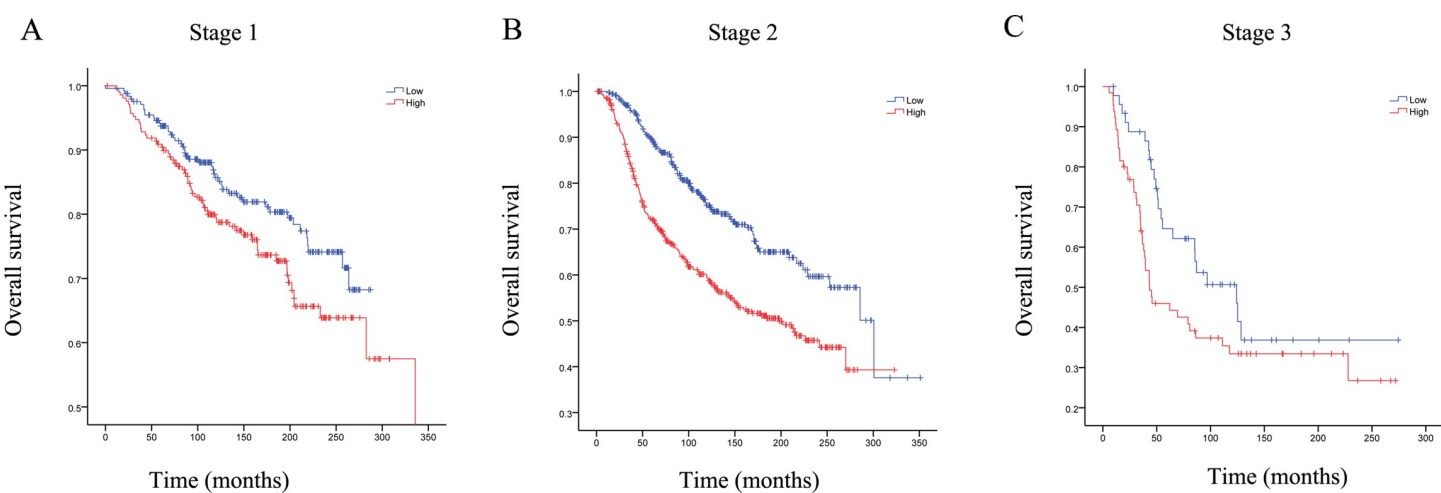

**Fig 7. The effects of RRM2 high expression and low expression on the overall survival of breast cancer patients.** Our study found that the RRM2 high expression group had a poor prognosis among stage1 and stage2 breast cancer patients. However, there was no significant difference in stage3. (A) stage 1; (B) stage 2; (C) stage 3.

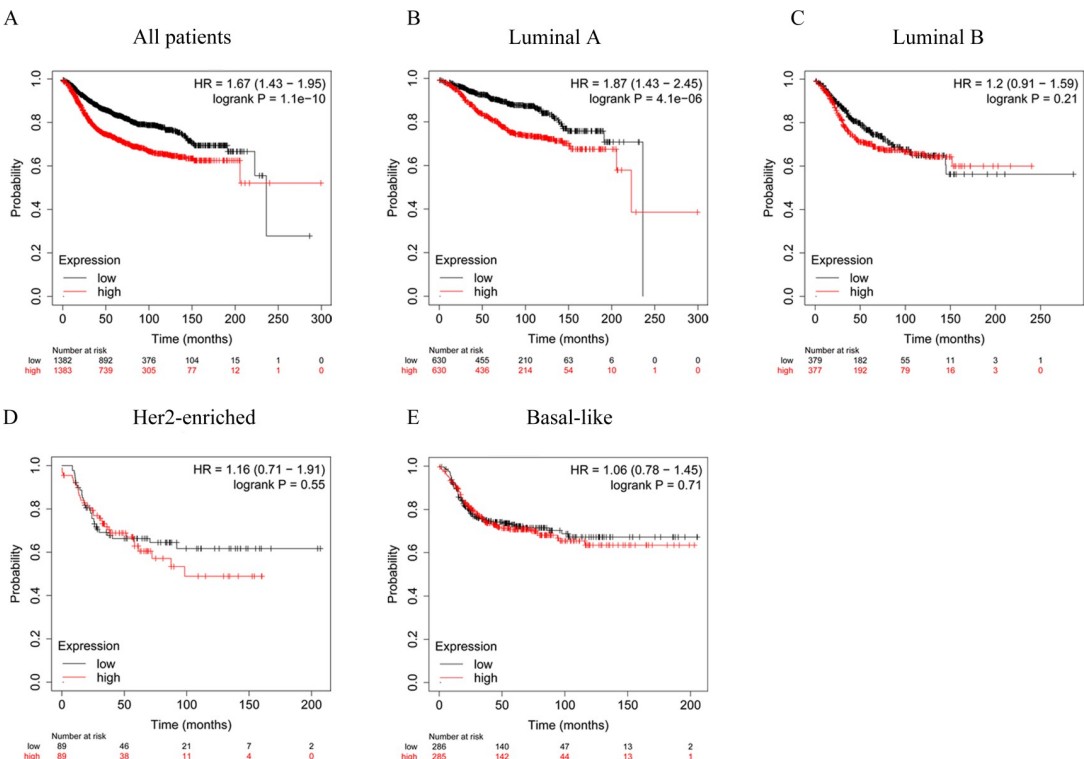

**Fig 8. The effects of RRM2 expression on the DMFS of breast cancer patients.** (A) All breast cancer patients, (B) Luminal A, (C) Luminal B, (D) Her-2 enriched, and (E) Basal-like.

found that RRM2 was highly expressed in lung adenocarcinoma, pancreatic adenocarcinoma, oral squamous cell carcinoma, and cervical cancer [5, 20–23].Our study found that relative levels of RRM2 mRNA expression were up-regulated in breast cancer tissues and cell lines. The

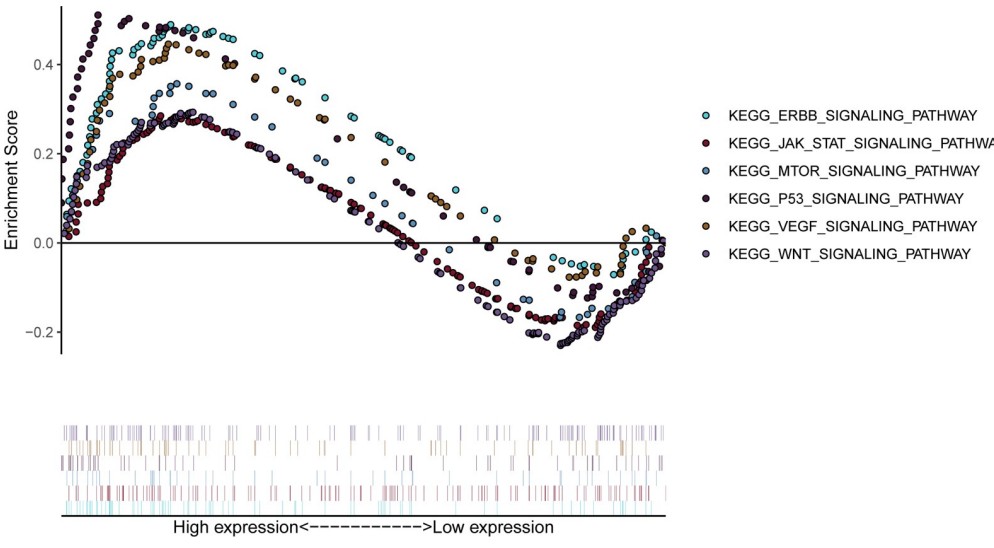

**Fig 9. GSEA analysis showed the KEGG signaling pathway in breast cancer patients with high expression of RRM2.**

same trend was observed when we examined the level of RRM2 protein expression, which is consistent with the RRM2 expression trend in other tumors. This finding suggests that differentially expressed RRM2 may affect the occurrence and development of breast cancer.

Guo Q et al. found that the differentially expressed genes are associated with the clinico-pathology of breast cancer [24–28]. Our study found that in 1,350 breast cancer patients, the expression of RRM2 was significantly associated with age, PAM50 classification, ER status, PR status, Her-2 status, menopause, or tumor grade (P<0.05). However, there were no significant differences in the Number of positive lymph nodes, Tumor size, and Tumor stage. This is the first study that reports the association of RRM2 expression and breast cancer. We infer that the high expression of RRM2 has an impact on the clinicopathology of breast cancer patients.

Univariate and multivariate Cox regression analysis showed that RRM2 can be used as an independent prognostic factor, which is significantly related to OS in patients with breast cancer. Furthermore, the number of positive lymph nodes, ER, Her-2 status, tumor size, and tumor stage can also be used as independent prognosis factors to predict the OS in breast cancer patients. Our study showed that, among the molecular subtypes of breast cancer, the Luminal A subtype had the best OS, and the basal-like subtype had the worst OS, which is consistent with the study by Blockhuys S et al. [29].

Mathias C et al. found that in different subtypes of breast cancer, differentially expressed genes can affect the OS of breast cancer patients [30–33]. To determine whether the high expression of RRM2 was associated with the molecular subtypes of breast cancer, we conducted KM survival analysis. High expression of RRM2 had poor OS in patients with Luminal A subtype and Normal-like subtype breast cancer. Many Chinese experts consider that Luminal A subtype breast cancer will achieve a "poor response to chemotherapy". The TAI-LORx study and the MINDACT study [34, 35] showed that in patients with Luminal type, the results of genetic testing can screen patients that should avoid chemotherapy. In addition, previous studies have pointed out that high expression of RRM2 may promote chemotherapy resistance in breast cancer patients [36, 37]; thus, in Luminal breast cancer, high expression of RRM2 may influence the OS of patients by promoting drug resistance. However, this assumption is only based on our inference and on the findings from a simple database sample analysis and existing literature, and the specific influencing factors still need to be explored in our follow-up research work. There was no significant difference in OS among Basal-like subtypes, Luminal B subtypes, Claudin low subtypes, and Her-2-enriched subtypes. Therefore, the high expression of RRM2 may only affect the OS of patients with specific subtypes of breast cancer.

We also evaluated the impact of high-expression of RRM2 in different stages of breast cancer patients on OS. The results showed that in stage 1 and stage 2 breast cancer patients, the high expression of RRM2 was associated with poor OS. The above results suggest that the high expression of RRM2 may have an impact on the OS of breast cancer patients of specific subtypes and specific stages.

Does the expression level of RRM2 affect the DMFS of breast cancer patients? We performed a KM online survival analysis and the results showed that, of all the breast cancer types, patients with high RRM2 expression and Luminal A breast cancer had the worse DMFS. This suggests that up-regulation of RRM2 expression not only affects the OS of breast cancer patients but also affects DMFS, especially for Luminal A subtype breast cancer, the specific mechanism still needs to be further explored. The study by Zhang Hang et al. [38]. revealed the prognostic role and therapeutic significance of RRM2 in estrogen receptor-negative breast cancer. Our study analyzed the prognostic role of highly expressed RRM2 in various breast cancer subtypes, and finally we focused on the prognostic role of RRM2 in Luminal A breast cancer—an estrogen receptor-positive subtype of breast cancer.

Finally, a GSEA analysis was performed. The results showed that the high expression of RRM2 was mainly enriched in the ERBB signaling pathway, JAK-STAT signaling pathway, mTOR signaling pathway, P53 signaling pathway, VEGF signaling pathway, and WNT signaling pathway. Ghaemi Z found that miR-326 can inhibit the occurrence of breast cancer by regulating the ERBB/PI3K pathway [39], and in addition, lncRNA PCAT7 can activate the ERBB/PI3K/AKT pathway to promote breast cancer progression [40]. Xu J et al. confirmed that the JAK-STAT signaling pathway was associated with the occurrence and development of breast cancer, bladder cancer, ovarian cancer and oral cancer [41–44]. Shorning et al. found that activation of the mTOR signaling pathway could lead to prostate cancer and breast cancer, disease progression, and was associated with treatment resistance [45, 46]. Ouyang LW et al. found that activation of the P53 signaling pathway could inhibit the growth of lung cancer and gastric cancer [47, 48]. Wang et al. found that Cystathionine-gamma-lyase can promote breast cancer metastasis through the VEGF signaling pathway [49]. Cui et al. found that up-regulated lncRNA SNHG1 can activate wnt-related signaling pathways to promote tumorigenesis [50]. Therefore, we further infer that the high expression of RRM2 may affect the progression and prognosis of breast cancer through the above signaling pathways.

This study has certain innovations. First, we confirmed that RRM2 is highly expressed in breast cancer tissues through multiple data sets (METABRIC, TCGA Agilent, TCGA RNA-seq, GSE5364). Instead of using the GEO database alone, we performed an analysis of the TCGA database and the METABRIC database, which have very large breast cancer tissue samples (over 2000 tissue samples). This provides a more reliable analytical basis for our subsequent analysis. Next we verified that the expression of RRM2 in breast cancer cell lines is up-regulated by qRT-PCR and western blotting, which increases the reliability of the data. Second, it is the first time that the highly expressed RRM2 has poor OS and DMFS in patients with breast cancer with specific molecular subtypes and specific stages.

Our research also has certain limitations. This study uses bioinformatics analysis methods to study breast cancer data in public databases. We only performed a few experiments to clarify the mRNA and protein expression levels of RRM2, therefore, the role and mechanism of RRM2 in breast cancer cells and breast cancer tissues need to be further verified and explored by our subsequent studies.

## Conclusion

Highly expressed RRM2 is associated with poor OS and DMFS of breast cancer patients. Furthermore, the high expression of RRM2 was associated with poor OS in breast cancer patients characterized by specific molecular subtypes (Luminal A subtype and Normal-like subtype) and specific stages (stage 1 and stage 2). RRM2 can be used as a biomarker, and is associated with the survival and prognosis of breast cancer patients.

## Supporting information

**S1 Text. Screened clinical information of breast cancer patients in METABRIC database.** (TXT)

**S1 File. Source codes used for bioinformatics analysis.** (XLS)

**S2 File. Datasets information in BCIP database.** (XLSX)

**S3 File. Table of statistics of RRM2 expression values in four datasets.** (XLSX)

**S4 File. RRM2 transcriptome expression data in METABRIC dataset.**
(XLSX)

**S5 File. RRM2 transcriptome expression data in TCGA Agilent dataset.**
(XLSX)

**S6 File. RRM2 transcriptome expression data in RNA-Seq dataset.**
(XLSX)

**S7 File. RRM2 transcriptome expression data in GSE5364 dataset.**
(XLSX)

## Acknowledgments

We thank Professor Yong Tang, Peng Qi, and Yan He for their support.

## Author Contributions

**Conceptualization:** Shen-chao Shi, Yi Zhang, Tao Wang.

**Data curation:** Shen-chao Shi, Yi Zhang.

**Formal analysis:** Tao Wang.

**Software:** Shen-chao Shi.

**Visualization:** Shen-chao Shi.

**Writing – original draft:** Shen-chao Shi, Yi Zhang.

**Writing – review & editing:** Yi Zhang, Tao Wang.

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
