## [Decision Letter · Decision Letter 0]

6 Dec 2021

PONE-D-21-23611High RRM2 expression has poor overall survival in specific types of breast cancerPLOS ONE

Dear Dr. Wang,

Thank you for submitting your manuscript to PLOS ONE. After careful consideration, we feel that it has merit but does not fully meet PLOS ONE’s publication criteria as it currently stands. Therefore, we invite you to submit a revised version of the manuscript that addresses the points raised by the reviewers below during the review process.

We look forward to receiving your revised manuscript.

Kind regards,

Surinder K. Batra

Academic Editor

PLOS ONE

Reviewers' comments:

Reviewer's Responses to Questions

**Comments to the Author**

1. Is the manuscript technically sound, and do the data support the conclusions?

Reviewer #1: Partly

Reviewer #2: Yes

2. Has the statistical analysis been performed appropriately and rigorously? 

Reviewer #1: Yes

Reviewer #2: Yes

3. Have the authors made all data underlying the findings in their manuscript fully available?

Reviewer #1: Yes

Reviewer #2: Yes

4. Is the manuscript presented in an intelligible fashion and written in standard English?

Reviewer #1: No

Reviewer #2: Yes

5. Review Comments to the Author

Reviewer #1: The manuscript entitled “High RRM2 expression has poor overall survival in specific types of breast cancer” by Shi et al, evaluates the role of RRM2 as a prognostic biomarker in breast cancer and attempts to relate the expression level of RRM2 in the METABRIC database with the overall survival of breast cancer patients. Although the authors claim that RRM2 has not been studied in detail in breast cancer, more comprehensive studies such as “Prognostic and therapeutic significance of ribonucleotide reductase small subunit M2 in estrogen-negative breast cancers” already exist. There is no mention of this study or any other existing literature on breast cancer. This substantially undermines the novelty of this study. Further, there are other concerns.

1.In the results section, more explanation on the type of correlation of RRM2 expression with clinical features is needed. Similarly, the authors need to explain the results of Figure 2, detailing the trends that are observed.

2.The figure legend in Figure 4B is misrepresented. Figure 4B shows that Stage 1 breast cancer patients had the best overall survival and stage 4 the worst and doesn’t relate the expression of RRM2 to poor overall survival in stage 1 and stage 2 breast cancer patients.

3.The manuscript provides no explanation for why high expression of RRM2 has poor overall survival in patients with Luminal A subtype and Normal-like subtype breast cancers.

4.The manuscript needs to be corrected for grammar and flow. For example, the sentences in lines 103 to 105 need to be simplified. Grammatical errors and language inconsistencies impair flow and render reading the manuscript a tedious task.

5.The last line of materials and methods in the abstract “Finally, using GSEA to study the differentially expressed RRM2 may affect the related pathways of breast cancer progression” needs to be rephrased to increase comprehension.

Unless the authors can support their findings with some experimental data, this manuscript is not suitable for publication.

Reviewer #2: This manuscript demonstrates the role of RRM2 in breast cancer. Authors have shown that higher RRM2 of RRM2 is associated with a worse prognosis in breast cancer patients. They utilized publically available datasets such as KM plotters via utilizing univariate or multivariate analyses and GSEA analysis to show that RRM2 is indeed an important molecule for breast cancer progression and its higher expression is associated with ErbB and other cancer-associated signaling pathways. Overall, these studies suggest that RRM2 could be used as a prognostic factor for breast cancer patients. Although the findings are interesting and well organized, these studies are lacking some further details. RRM2 has been associated with drug resistance in other cancer types. Is RRM2 involved with chemoresistance in several breast cancer subtypes? It is also important to know if RRM2 is associated with distant metastasis-free survival. In addition, this reviewer would like to know if the gene expression data of RRM2 is also correlated with the proteomic data or protein atlas data.

6. PLOS authors have the option to publish the peer review history of their article (what does this mean?). If published, this will include your full peer review and any attached files.

Reviewer #1: No

Reviewer #2: No

---

## [Author Response · Author response to Decision Letter 0]

20 Jan 2022

Dear editor

Thank you for your letter and the reviewer's comments concerning our manuscript entitled " High RRM2 expression has poor overall survival in specific types of breast cancer"( Manuscript Number: PONE-D-21-23611). Those comments are valuable and very helpful. We have read through the comments carefully and have made corrections. Based on the instructions provided in your letter, we uploaded the file of the revised manuscript. The responses to the reviewer's comments are presented following.

Sincerely.

Tao Wang

Reviewer #1: The manuscript entitled “High RRM2 expression has poor overall survival in specific types of breast cancer” by Shi et al, evaluates the role of RRM2 as a prognostic biomarker in breast cancer and attempts to relate the expression level of RRM2 in the METABRIC database with the overall survival of breast cancer patients. Although the authors claim that RRM2 has not been studied in detail in breast cancer, more comprehensive studies such as “Prognostic and therapeutic significance of ribonucleotide reductase small subunit M2 in estrogen-negative breast cancers” already exist. There is no mention of this study or any other existing literature on breast cancer. This substantially undermines the novelty of this study. Further, there are other concerns.

Response：

(1)We agree with your review comments. Our study does have certain limitations, such as lack of novelty and lack of more in-depth mechanism research. But we still believe that this study has a certain value and can provide some new ideas and perspectives for breast cancer research. (2) The study“Prognostic and therapeutic significance of ribonucleotide reductase small subunit M2 in estrogen-negative breast cancers”by Zhang Hang et al.[1] revealed the prognostic role and therapeutic significance of RRM2 in estrogen receptor-negative breast cancer. Our study analyzed the prognostic role of highly expressed RRM2 in various breast cancer subtypes, and finally, we focused on the prognostic role of RRM2 in estrogen receptor-positive breast cancer. We mentioned this research in the Discussion section(line 466-471). (3)Instead of using the GEO database alone, we performed an analysis of the TCGA database and the METABRIC database, which have very large breast cancer tissue samples (over 2000 breast cancer samples). This provides a more reliable analytical basis for our subsequent analysis. (4) In the study of Zhang Hang et al.[1], they analyzed 4 breast cancer subtypes, and our study also analyzed the claudin-low subtype in addition to the above 4 breast cancer types, and the data will be more comprehensive.

Q1.In the results section, more explanation on the type of correlation of RRM2 expression with clinical features is needed. Similarly, the authors need to explain the results of Figure 2, detailing the trends that are observed.

Response: We have explained the relationship between RRM2 and clinical features in detail following the reviewer comments.(line270-281).

Q2.The figure legend in Figure 4B is misrepresented. Figure 4B shows that Stage 1 breast cancer patients had the best overall survival and stage 4 the worst and doesn’t relate the expression of RRM2 to poor overall survival in stage 1 and stage 2 breast cancer patients.

Response: Thank you very much for reminding us that the order of our previous legends was wrong and has been corrected.(line326-334)

Q3.The manuscript provides no explanation for why high expression of RRM2 has poor overall survival in patients with Luminal A subtype and Normal-like subtype breast cancers.

Response: Many Chinese experts consider luminal A subtype breast cancer "poor response to chemotherapy". The TAILORx study and the MINDACT study showed that in patients with Luminal type, the results of genetic testing can screen some patients to avoid chemotherapy[2,3]. In addition, some literature pointed out that high expression of RRM2 may promote drug resistance in breast cancer patients[4,5], so we infer that in Luminal breast cancer, high expression of RRM2 may affect the overall survival rate of patients by promoting the generation of drug resistance. Of course, this assumption is only based on our inference based on database sample analysis and existing literature, and the specific influencing factors still need to be explored in our follow-up research work.(line437-449)

Q4.The manuscript needs to be corrected for grammar and flow. For example, the sentences in lines 103 to 105 need to be simplified. Grammatical errors and language inconsistencies impair flow and render reading the manuscript a tedious task.

Response: We have linguistically polished previous manuscripts for grammar, logic, fluency, clarity, and accuracy. We hope that the readability of the revised manuscript can be improved.

Q5.The last line of materials and methods in the abstract “Finally, using GSEA to study the differentially expressed RRM2 may affect the related pathways of breast cancer progression” needs to be rephrased to increase comprehension.

Response: We have made changes in the manuscript in response to your comments.(line 37-40)

Unless the authors can support their findings with some experimental data, this manuscript is not suitable for publication.

Response: Based on your valuable comments, we have added some experimental data, such as the expression level of RRM2 in breast cancer patient tissue samples and breast cancer cell lines through western blot experiments, which verified the bioinformatics data in this study.(line246-254)

References

[1] Zhang H, Chu P, Zheng S, Yen Y. Prognostic and therapeutic significance of ribonucleotide reductase small subunit M2 in estrogen-negative breast cancers. BMC Cancer. 2014 Sep 11;14:664. 

[2] Sparano JA, Gray RJ, Makower DF, Sledge GW Jr. Adjuvant Chemotherapy Guided by a 21-Gene Expression Assay in Breast Cancer. N Engl J Med. 2018 Jul 12;379(2):111-121. 

[3] Cardoso F, van't Veer LJ, Bogaerts J, Piccart M; MINDACT Investigators. 70-Gene Signature as an Aid to Treatment Decisions in Early-Stage Breast Cancer. N Engl J Med. 2016 Aug 25;375(8):717-29. 

[4] Putluri N, Maity S, Kommagani R, Sreekumar A. Pathway-centric integrative analysis identifies RRM2 as a prognostic marker in breast cancer associated with poor survival and tamoxifen resistance. Neoplasia. 2014 May;16(5):390-402. 

[5] Zhan Y, Jiang L, Jin X, Qiu Y. Inhibiting RRM2 to enhance the anticancer activity of chemotherapy. Biomed Pharmacother. 2021 Jan;133:110996. 

Reviewer #2: This manuscript demonstrates the role of RRM2 in breast cancer. Authors have shown that higher RRM2 of RRM2 is associated with a worse prognosis in breast cancer patients. They utilized publically available datasets such as KM plotters via utilizing univariate or multivariate analyses and GSEA analysis to show that RRM2 is indeed an important molecule for breast cancer progression and its higher expression is associated with ErbB and other cancer-associated signaling pathways. Overall, these studies suggest that RRM2 could be used as a prognostic factor for breast cancer patients. Although the findings are interesting and well organized, these studies are lacking some further details. RRM2 has been associated with drug resistance in other cancer types. Is RRM2 involved with chemoresistance in several breast cancer subtypes? It is also important to know if RRM2 is associated with distant metastasis-free survival. In addition, this reviewer would like to know if the gene expression data of RRM2 is also correlated with the proteomic data or protein atlas data.

Response: 

To evaluate the effect of RRM2 expression level on distant metastasis-free survival (DMFS) in breast cancer patients, we performed Kaplan-Meier online survival analysis, and the results showed that, among all breast cancer patients, the RRM2 high expression group had worse DMFS (P < 0.05). We further evaluated the effect of RRM2 expression on DMFS in patients with different subtypes of breast cancer, and the results showed that in Luminal A breast cancer patients, high expression of RRM2 had worse DMFS (P < 0.05). However, The expression level of RRM2 has no significant correlation with the Basal-like subtype, the Luminal B subtype, basal-like subtype, and Her-2 enriched subtype. As shown in Fig 8.(line373-389)

We have confirmed that the relative mRNA expression level of RRM2 is up-regulated in breast cancer tissues and breast cancer cell lines by qRT-PCR and bioinformatics analysis. Next, we also hope to clarify whether there is a certain correlation between the RRM2 gene expression level and the protein expression level. We searched the Human Protein Atlas (HPA database) https://www.proteinatlas.org/, unfortunately, there are limited immunohistochemical data on RRM2 in breast cancer tissues. Then we detected the protein expression level of RRM2 by Western blot, and the results showed that the protein expression of RRM2 was significantly up-regulated in breast cancer tissues compared with adjacent normal tissues. At the same time, we also detected the protein expression level of RRM2 in breast cancer cell lines, and the same trend is obtained (Fig 2B-2C). Of course, more about RRM2 proteomics and protein changes in vital pathways in breast cancer need to be further explored.(line246-266)

RRM2 has been associated with drug resistance in other cancer types. Is RRM2 involved with chemoresistance in several breast cancer subtypes? In response to the above questions, we conducted a literature search. At present, the research of scholars mainly focuses on tamoxifen resistance (an endocrine therapy drug for breast cancer), and there is almost no research on chemoresistance in specific breast cancer subtypes. . We are very grateful for the opinions of the reviewers, which also provide a good direction and research ideas for our follow-up research. We need to design a feasible research protocol to further explore whether RRM2 affects chemoresistance in specific breast cancer subtypes.

---

## [Decision Letter · Decision Letter 1]

28 Feb 2022

High RRM2 expression has poor prognosis in specific types of breast cancer

PONE-D-21-23611R1

Dear Dr. Wang,

We’re pleased to inform you that your manuscript has been judged scientifically suitable for publication and will be formally accepted for publication once it meets all outstanding technical requirements.

Kind regards,

Surinder K. Batra

Academic Editor

PLOS ONE

Additional Editor Comments (optional):

Reviewers' comments:

Reviewer's Responses to Questions

**Comments to the Author**

1. If the authors have adequately addressed your comments raised in a previous round of review and you feel that this manuscript is now acceptable for publication, you may indicate that here to bypass the “Comments to the Author” section, enter your conflict of interest statement in the “Confidential to Editor” section, and submit your "Accept" recommendation.

Reviewer #1: All comments have been addressed

Reviewer #2: All comments have been addressed

2. Is the manuscript technically sound, and do the data support the conclusions?

Reviewer #1: Yes

Reviewer #2: Yes

3. Has the statistical analysis been performed appropriately and rigorously? 

Reviewer #1: Yes

Reviewer #2: Yes

4. Have the authors made all data underlying the findings in their manuscript fully available?

Reviewer #1: Yes

Reviewer #2: Yes

5. Is the manuscript presented in an intelligible fashion and written in standard English?

Reviewer #1: Yes

Reviewer #2: Yes

6. Review Comments to the Author

Reviewer #1: (No Response)

Reviewer #2: I appreciate authors for including new data to answer my queries. After revision, this manuscript is improved significantly.

7. PLOS authors have the option to publish the peer review history of their article (what does this mean?). If published, this will include your full peer review and any attached files.

Reviewer #1: No

Reviewer #2: No

---

## [Editor Report · Acceptance letter]

4 Mar 2022

PONE-D-21-23611R1 

High RRM2 expression has poor prognosis in specific types of breast cancer 

Dear Dr. Wang:

I'm pleased to inform you that your manuscript has been deemed suitable for publication in PLOS ONE. Congratulations! Your manuscript is now with our production department. 

Kind regards, 

on behalf of

Prof. Surinder K. Batra 

Academic Editor

PLOS ONE